# Associations between PM_2.5_ Components and Mortality of Ischemic Stroke, Chronic Obstructive Pulmonary Disease and Diabetes in Beijing, China

**DOI:** 10.3390/toxics12060381

**Published:** 2024-05-23

**Authors:** Hao Feng, Yisen Yang, Hong Ye, Jing Xu, Meiduo Zhao, Ye Jin, Shuyang Zhang

**Affiliations:** 1Department of Cardiology, State Key Laboratory of Complex Severe and Rare Diseases, Peking Union Medical College Hospital, Chinese Academy of Medical Sciences & Peking Union Medical College, Beijing 100730, China; haocams@163.com; 2Department of Epidemiology and Biostatistics, Institute of Basic Medical Sciences Chinese Academy of Medical Sciences, School of Basic Medicine Peking Union Medical College, Beijing 100005, China; yyiyis@163.com (Y.Y.); yehong_0703@163.com (H.Y.); xujing_inbj@163.com (J.X.); meiduo_zhao@126.com (M.Z.); 3Center of Environmental and Health Sciences, Chinese Academy of Medical Sciences & Peking Union Medical College, Beijing 100005, China; 4Center for Digital Medicine and Artificial Intelligence, Institute of Clinical Medicine, Peking Union Medical College Hospital, Beijing 100730, China; 5Department of Rare Diseases, Peking Union Medical College Hospital, Chinese Academy of Medical Sciences & Peking Union Medical College, Beijing 100730, China

**Keywords:** mortality, air pollutants, ischemic stroke, chronic obstructive pulmonary disease, diabetes, distributed lag nonlinear model

## Abstract

Ischemic stroke (IS), chronic obstructive pulmonary disease (COPD) and diabetes mellitus (DM) account for a large burden of premature deaths. However, few studies have investigated the associations between fine particular matter (PM_2.5_) components and mortality of IS, COPD and DM. We aimed to examine these associations in Beijing, China. Data on daily mortality, air pollutants and meteorological factors from 2008 to 2011 in Beijing were collected. Daily concentrations of five PM_2.5_ components, namely, sulfate ion (SO_4_^2−^), ammonium ion (NH_4_^+^), nitrate ion (NO_3_^−^), organic matter (OM) and black carbon (BC), were obtained from the Tracking Air Pollution (TAP) database in China. The association between PM_2.5_ components and daily deaths was explored using a quasi-Poisson regression with the distributed lag nonlinear model (DLNM). The average daily concentrations of SO_4_^2−^, NH_4_^+^, NO_3_^−^, OM and BC were 11.24, 8.37, 12.00, 17.34 and 3.32 μg/m^3^, respectively. After adjusting for temperature, relative humidity, pressure, particulate matter less than 10 μm in aerodynamic diameter (PM_10_), nitrogen dioxide (NO_2_) and sulfur dioxide (SO_2_), an IQR increase in OM at lag day 2 and lag day 6 was associated with an increased DM mortality risk (RR 1.038; 95% CI: 1.005–1.071) and COPD mortality risk (RR 1.013; 95% CI: 1.001–1.026). An IQR increase in BC at lag day 0 and lag day 6 was associated with increased COPD mortality risk (RR 1.228; 95% CI: 1.017–1.48, RR 1.059; 95% CI: 1.001–1.121). Cumulative exposure to SO_4_^2−^ and NH_4_^+^ was associated with an increased mortality risk for IS, with the highest effect found for lag of 0–7 days (RR 1.085; 95% CI: 1.010–1.167, RR 1.083; 95% CI: 1.003–1.169). These effects varied by sex and age group. This study demonstrated associations of short-term exposure to PM_2.5_ components with increased risk of IS, COPD and DM mortality in the general population. Our study also highlighted susceptible subgroups.

## 1. Introduction

Each year, an average of 41 million people die from noncommunicable diseases (NCDs), accounting for 74% of all deaths worldwide [1]. Cardiovascular diseases, cancers, chronic respiratory diseases and diabetes are the four main types of NCD and account for over 80% of all premature NCD deaths [1]. In China, these four main NCDs account for 88.5% of total deaths [2]. Among cardiovascular diseases, stroke leads to nearly 2.5 million new cases and 1.6 million deaths every year in China [3]. Similar to other countries, ischemic stroke (IS) is the most common type of stroke in China, accounting for 80% of all strokes [3]. Chronic obstructive pulmonary disease (COPD) was the third leading cause of death in China and accounted for more than 0.9 million deaths in 2013 [4]. China had the largest number of patients with diabetes mellitus (DM) in the world in 2013 and reported a rapid increase in DM prevalence and mortality from 2005 to 2020 [5,6]. Due to the increasing burden from IS, COPD and DM, it is crucial to identify key risk factors to prevent premature deaths from these NCDs.

Air pollution was responsible for 6.7 million deaths in 2019 globally [7]. Among air pollutants, fine particulate matter with an aerodynamic diameter less than or equal to 2.5 μm (PM_2.5_) is considered more hazardous due to its smaller size, which means that it can reach deeply into the respiratory tract and enter the bloodstream, and its large surface area, which can carry various toxic chemicals [8]. Air pollution has become a significant public health threat in China as a result of industrialization and urbanization in the past few decades.

Increasing evidence has shown that major air pollutants, such as PM_2.5_, particulate matter less than 10 μm in aerodynamic diameter (PM_10_), nitrogen dioxide (NO_2_) and sulfur dioxide (SO_2_), carbon monoxide (CO) and ozone (O_3_), are closely associated with increased risk of morbidity and mortality [9,10,11,12]. Especially for PM_2.5_, there is growing and consistent evidence that it leads to a sustained increase in mortality from cardiovascular and respiratory diseases [13,14]. For instance, a 10 μg/m^3^ increase in PM_2.5_ concentration was associated with a 2.7% increase in respiratory disease mortality [15]. Studies have also shown that the associations between PM_2.5_ mass concentrations and health outcomes vary in magnitude by region, which could be due to differences in PM_2.5_ sources and chemical composition [14,16]. PM_2.5_ components also varied across time in the same region [17]. For example, in Beijing, secondary inorganic aerosol (sulfate ion (SO_4_^2−^), ammonium ion (NH_4_^+^), nitrate ion (NO_3_^−^)) have increased significantly and become the major components since 2009 [17]. However, the complex composition of PM_2.5_ has made it difficult to quantify the most harmful constituents of PM_2.5_. Although some studies have shown that different components of PM_2.5_ have different health effects [18], most of the existing studies have investigated PM_2.5_ mass as a whole.

Most of the previous studies on the acute effects of PM_2.5_ components were conducted in Europe and the USA [14]. Such studies were limited in other regions including China, which could be due to the concentration estimate of these components. There are limited studies investigating the associations between PM_2.5_ components and mortality from IS, COPD and DM [8,19,20]. It is therefore still unclear which PM_2.5_ component is more harmful due to the lack of studies and inconsistent findings [21]. There are currently no guidelines on the chemical composition of PM_2.5_. Investigating the relationships between PM_2.5_ components and mortality could help expand the current World Health Organization (WHO) air quality guidelines to a more detailed level and promote more targeted regulations according to emission sources.

In addition to investigating the associations between PM_2.5_ components and mortality, exploring their exposure–response relationships can provide a full picture of the impacts of these components on mortality. To our knowledge, there is a lack of evidence from existing studies that explore this relationship. Therefore, in this study, we examined the associations between five major PM_2.5_ components, namely, SO_4_^2−^, NH_4_^+^, NO_3_^−^, organic matter (OM) and black carbon (BC), and mortality risk from IS, COPD and DM in a population of 19.2 million in Beijing, China. We also investigated the exposure–response relationships in the whole population and subgroups. We further explored whether the magnitude of effects differs by sex and age.

## 2. Materials and Methods

### 2.1. Study Site

Beijing (east longitude: 115.4°–117.5°, north latitude: 39.4°–41.1°), the capital of China, is located in a plain area with a resident population of 19.2 million during the study period (from 1 January 2008 to 30 December 2011). It is the key area of the Air Pollution Prevention and Management Plan for the Beijing–Tianjin–Hebei regions.

### 2.2. Data Collection

#### 2.2.1. Exposure Measurement

Daily concentrations of five PM_2.5_ components, namely, SO_4_^2−^, NH_4_^+^, NO_3_^−^, OM and BC [22], at 10 km × 10 km spatial resolution in Beijing, China were obtained from the Tracking Air Pollution (TAP) database in China (http://tapdata.org.cn, accessed on 14 August 2023) [23]. These five chemical components of PM_2.5_ were chosen based on the evidence that they are the leading components of PM_2.5_. From the observation data, the concentrations of SO_4_^2−^, NH_4_^+^, NO_3_^−^, OM and BC accounted for 82.5% of the total PM_2.5_ concentration [24].

TAP estimated near-real-time PM_2.5_ components data in China from 2000 using Weather Research and Forecasting (v3.9.1)-Community Multiscale Air Quality (v5.2) chemical transport models by combining data from ground observations from ground-based monitors and predictors (i.e., satellite-based aerosol optical depth, bottom-up emission inventory, meteorological fields, land use, population and elevation) [23,24]. The TAP estimates for PM_2.5_ components showed good consistency with the ground measurements, with the cross-validation correlation coefficient (r)s being 0.70 for SO_4_^2−^, 0.75 for NH_4_^+^, 0.75 for NO_3_^−^, 0.72 for OM and 0.64 for BC [23]. We selected 7 days as the maximum lag length in our models. Additionally, to capture the short- and long-term effects, in our sensitivity analysis, we selected 4 and 14 days as the maximum lag days based on previous studies on PM_2.5_ and mortality [14,25,26].

#### 2.2.2. Outcome

Daily mortality data were obtained from the Causes of Death Registry of Chinese Centre for the Disease Control and Prevention (China CDC). The causes of death were recorded by the primary diagnosis coded according to the International Disease Classification Codes, version 10 (ICD-10) https://icd.who.int/browse10/2019/en accessed on 14 August 2023. Three cause-specific deaths were included in this study: IS (codes I63), DM (codes E10-E14) and COPD (codes J41-J44).The mortality data also include the individual’s age and sex.

#### 2.2.3. Covariates

To control for the effect of meteorological factors on mortality, we collected daily mean temperature, relative humidity and pressure from the local meteorological administration in Beijing. We also collected data on three pollutants, PM_10_, NO_2_ and SO_2_, from 12 National Air Quality Monitor (NAQM) stations in Beijing, because previous studies have found associations between these pollutants and death [27,28].

### 2.3. Statistical Analysis

Descriptive analyses of meteorological factors, particulate matter fractions and daily deaths of the population were performed. The indicators included mean and standard deviation (SD), mean absolute deviation (MAD), minimum (Min), 25th Percentile (P_25_), 50th Percentile (P_50_), 75th Percentile (P_75_), interquartile range (IQR) and maximum (Max). Spearman’s correlation coefficient was used to test the correlation between PM_2.5_ components and other air pollutants.

We explored the association between particulate matter fractions and daily deaths using a quasi-Poisson regression with the distributed lag nonlinear model (DLNM).
$$Y_{t}=QuasiPoisson\;\left(\mu_{t}\right)$$
$$\log\left(\mu_{t}\right)=\alpha +cb\left(x,\;lag\right)+cb\left(temp\right)+ns\left(time,\;df\right)+\beta DOW_{t}+COVs$$
where Y_t_ denotes the number of cause-specific deaths on day *t*; *α* denotes the intercept; *cb* is the cross-basis function; *x* is the component of PM_2.5_ we are interested in, and the effect of temperature also enters the model as a cross-basis term; *ns* (*time*, *df*) denotes a term of natural cubic spline of time with 7 degrees of freedom per year to account for unmeasured time-dependent confounders, long term trends, and seasonality; *DOW*, a categorical variable, is used to control for day-of-week effects; *COVs* denotes other covariates, and in this study, we included PM_10_, NO_2_ and SO_2_, as well as relative humidity and atmospheric pressure, all entered into the model as natural cubic splines. We utilize quasi-Akaike information criteria (qAIC) to screen for the best combination of degrees of freedom. DLNM can simultaneously explore the lag–response and exposure–response for each PM_2.5_ component and mortality of IS, COPD and DM [29,30].

#### 2.3.1. Calculation of Attributable Fractions and Numbers

We further calculated the attributable fractions between PM_2.5_ components and mortality. The attributable fractions and numbers were calculated using methods described in a previous study [31]. Attributable numbers (AN) and attributable fractions (AF) are statistical measures used to quantify the impact of risk factors on health outcomes in a population.

AN refers to the absolute number of cases of a disease or health outcome that can be directly attributed to a specific risk factor. Essentially, it is the difference in the total number of cases that would occur with the risk factor versus without it.

We calculated it using the following formula:

AN = (Incidence in the exposed group−Incidence in the unexposed group) × Population size exposed.

AF, also known as attributable risks, provides a proportion or percentage of all cases of a certain outcome in a population that are attributable to a particular risk factor.

We calculated it using the following formula:$$\mathrm{AF}=\frac{\mathrm{Incidence}\;\mathrm{in}\;\mathrm{the}\;\mathrm{total}\;\mathrm{population}\;-\;\mathrm{Incidence}\;\mathrm{in}\;\mathrm{the}\;\mathrm{unexposed}\;\mathrm{group}}{\mathrm{Incidence}\;\mathrm{in}\;\mathrm{the}\;\mathrm{total}\;\mathrm{population}}\times 100\%$$

#### 2.3.2. Sensitivity Analysis

We conducted stratified analyses to evaluate whether the association differed in subgroups between male and female, old (≥65 years old) and young (<65 years old) populations. We further performed sensitivity analyses to test the robustness of our results by changing the maximum lag days (4 and 14 lag days) for all models. All analyses were performed using R software (v4.2.2, R Foundation for Statistical Computing, Vienna, Austria) with the packages “tidyverse”, “dlnm”, “splines”, “mgcv” and “FluMoDL” [31]. Two-tailed *p* < 0.05 was considered statistically significant.

This study was approved by the Institutional Review Board of the Institute of Basic Medical Sciences, Chinese Academy of Medical Sciences.

## 3. Results

### 3.1. Descriptive Statistics

The average daily deaths from IS, COPD and DM in Beijing between 1 January 2008 and 30 December 2011 were 15, 9 and 5 per day, respectively (Table 1). During the study period, the average daily concentrations of SO_4_^2−^, NH_4_^+^, NO_3_^−^, OM and BC were 11.24, 8.37, 12.00, 17.34 and 3.32 μg/m^3^, respectively. The mean daily concentrations of NO_2_, SO_2_ and PM_10_ were 49.82, 32.27 and 117.75 μg/m^3^. The mean daily temperature, relative humidity and barometric pressure were 13.21 °C, 50.87% and 1012 hPa. The daily concentrations of air pollutants and meteorological factors showed seasonal patterns with higher air pollution in winter (Appendix A).

The Spearman’s correlation coefficients between PM_2.5_ components (SO_4_^2−^, NH_4_^+^, NO_3_^−^, OM and BC) and other air pollutants (NO_2_, SO_2_ and PM_10_) are shown in Figure 1, indicating a close positive correlation between PM_2.5_ components and other major air pollutants (all *p* < 0.01). The highest correlation was between NH_4_^+^ and NO_3_^−^ (correlation coefficient = 0.976).

### 3.2. Associations between PM_2.5_ Components and Cause-Specific Mortality

The results of single-day lag (0–7) and cumulative lags (1–7) models on the relationship between PM_2.5_ components and cause-specific deaths for each IQR increase in average pollutant concentration are presented in Figure 2 and Appendix A. After adjusting for temperature, relative humidity, pressure, NO_2_, SO_2_ and PM_10_, an IQR increase in OM at lag day 2 and lag day 6 was associated with an increased DM mortality risk (RR 1.038, 95% CI: 1.005–1.071) and COPD mortality risk (RR 1.013, 95% CI: 1.001–1.026). An IQR increase in BC at lag day 0 and lag day 6 was associated with increased COPD mortality risk (RR 1.228, 95% CI: 1.017–1.482; RR 1.059, 95% CI: 1.001–1.121). There is no evidence of IS, COPD and DM mortality and other exposure after a single-day lag.

The estimated effects from the cumulative lag models were generally larger than the single-day lag models for all studied PM_2.5_ components on mortality from IS, COPD and DM. Cumulative exposure to SO_4_^2−^ and NH_4_^+^ was associated with an increased mortality risk for IS, with the highest effect found for lag 7 (RR 1.085, 95% CI: 1.010–1.167; RR 1.083, 95% CI: 1.003–1.169). Cumulative exposure to OM and BC was associated with increased mortality risks for COPD and DM. The cumulative exposures (from lag 1 to lag 7) showed an increasing trend for mortality from IS and COPD.

### 3.3. Exposure–response Relationships

In the whole population, the exposure–response curves for these five PM_2.5_ components with IS, COPD and DM mortality were approximately linear without thresholds (Figure 3). With the increase in concentrations for SO_4_^2−^, NH_4_^+^, NO_3_^−^, OM and BC, there was an increasing trend of relative risks of mortality from IS, COPD and DM. The exposure–response curves for different lag days and among subgroups are shown in the Appendix A (Appendix A).

### 3.4. Attributable Fractions (AF) and Numbers (AN)

The attributable fractions and numbers of IS, COPD and DM deaths attributable to SO_4_^2−^, NH_4_^+^, NO_3_^−^, OM and BC cumulative exposure across 7 lag days are listed in Table 2. For IS mortality, after a lag of 7 days, the number of deaths attributable to SO_4_^2−^ was 1597, accounting for 7.36% (95% CI: 1.03–13.59) of the deaths. For COPD, the number of deaths attributable to OM and BC was 1528 and 1611, corresponding to 11.39% (95% CI: 1.80–19.65) and 12.01% (95% CI: 2.16–20.42) of the deaths. There is no evidence of a significant finding that deaths from DM were attributable to these air pollutants.

### 3.5. Stratified Analysis

The stratified results showed that males were more susceptible to OM and BC exposure across up to 4 lag days for DM mortality (Appendix A), while females were more susceptible to OM and BC exposure across up to 7 lag days for COPD mortality. There is little evidence of IS, COPD and DM mortality and SO_4_^2−^, NH_4_^+^, NO_3_^−^, OM and BC exposure after a single-day lag.

After being stratified by age, for the elderly population (age ≥ 65 years old), exposure to higher levels of OM and BC was associated with an increased risk of COPD mortality (Appendix A). For the younger population (age < 65 years old), exposure to higher levels of OM and BC was associated with an increased risk of DM mortality. The highest relative risk was found among the younger population (age < 65 years old) for DM mortality from the cumulative 7 lag days of BC (RR 3.167, 95% CI: 1.467–6.847).

### 3.6. Sensitivity Analysis

When we changed the maximum lag days (lag 4 and 14 days) for the air pollutants, the estimated relative risks for DM and COPD mortality did not change substantially (Appendix A). In the 14 lag day models, we further found evidence of associations between exposure to NO_3_^−^ and an increased risk of IS mortality.

## 4. Discussion

To our knowledge, this is the first study to explore the associations between PM_2.5_ components and IS, COPD and DM mortality in China. Our results showed that higher exposure to PM_2.5_ components was associated with an increased risk of IS, COPD and DM mortality, especially for the cumulative exposure of multi-lag days. The effects on mortality risk varied by the specific component and the specific cause of death. For instance, cumulative higher exposure to SO_4_^2−^ and NH_4_^+^ was associated with an increased risk of IS mortality, while cumulative higher exposure to OM and BC was associated with an increased risk of COPD and DM mortality. These effects also varied by sex and age group. Higher exposure to OM and BC was associated with increased DM mortality risk in males and the younger population (age < 65 years old) but was associated with increased COPD mortality risk in females and the elderly population (age ≥ 65 years old). These short-term effects of PM_2.5_ components found in this study provided evidence of the difference found in studies investigating PM_2.5_ as a whole.

We found positive trends between PM_2.5_ components and IS mortality for cumulative multiple lag days, of which significant cumulative lag effects were found for SO_4_^2−^ and NH_4_^+^. This finding is consistent with previous studies. For example, a prospective cohort study of 90,672 adults in China from 2010 to 2017 found that the hazard ratio (HR) of IS mortality was 1.43 (95% CI: 1.00–2.84) and 2.13 (95% CI: 1.27–4.58) per 10 μg/m^3^ increase in SO_4_^2−^ and NH_4_^+^ [32]. Additionally, this study also found significant positive associations between other components (i.e., BC, OM and NO_3_^−^) and IS mortality. Another prospective cohort study from western Germany showed that SO_4_^2−^ and NH_4_^+^ have stronger effects on the incidence of cardiovascular events including stroke than other particulate matter components [33]. A study conducted among hospitalized patients with stroke from four provinces in China also demonstrated that BC, OM, SO_4_^2−^ and NO_3_^−^ were associated with stroke fatality [34]. A recent review on PM_2.5_ components and IS summarized that the possible mechanisms for IS by inorganic ions including SO_4_^2−^, NH_4_^+^, and NO_3_^−^ involve chronic inflammation, oxidative stress, platelet aggregation and vascular endothelial cell injury [8]. A systematic review concluded that short-term periods of exposure to BC and OM were more detrimental to cardiovascular mortality than other components [18]. Although our study is consistent with previous studies in establishing that PM_2.5_ components have detrimental effects on mortality, it is still inconclusive which components are more harmful.

In our study, we found that OM and BC were consistently associated with COPD mortality. The effects not only appeared immediately but were also delayed after exposure. For instance, we found significant associations between COPD mortality and BC concentrations at lag days 0 and 6, and a cumulative relative risk up to 1.46 (95% CI: 1.08–1.96) for lag 7. Similarly, we found the lag-specific and cumulative lag effects of OM on COPD deaths. The potential mechanisms of COPD by these PM_2.5_ components are not fully understood, but they may include impairing airway and lung functions by promoting inflammation and oxidative stress [20]. There were limited studies assessing the associations between PM_2.5_ components and COPD mortality that are directly comparable to our study. Respiratory mortality could be a good surrogate to compare considering that COPD accounted for 83% of deaths from chronic respiratory diseases [35,36]. A time-series analysis in central London between 2000 and 2005 found that SO_4_^2−^ and NO_3_^−^ were important for respiratory outcomes but there was no evidence of significant associations with respiratory mortality [37]. A systematic review of acute effects of PM_2.5_ components on mortality found a 1.10% (95% CI: 0.60–1.50) increase in respiratory mortality per 10 μg/m^3^ increase in black smoke [14]. However, there was no evidence of significant associations between other PM_2.5_ components and respiratory mortality. A more recent systematic review found that organic matter was associated with respiratory mortality [18].

BC and OM concentrations at cumulative multiday lag were also associated with cumulative relative risk for DM mortality. We found a relative risk of 1.46 (95% CI: 1.12–1.90) for DM mortality per IQR increase in BC at lag 0–2. Evidence from a population-based cohort of 69,210 adults in China showed that BC (OR 1.07, 95% CI: 1.01–1.15) and OM (OR 1.09, 95% CI: 1.02–1.16) were positively associated with diabetes [19]. A prospective cohort analysis of 2.1 million adults from Canada found a 10 μg/m^3^ elevation in PM_2.5_ exposure was associated with an increase in risk for diabetes mortality (HR 1.49, 95% CI: 1.37–1.62) [38]. A cohort study of US veterans also found that PM_2.5_ exposure was associated with diabetes mortality [9]. There is a lack of studies on PM_2.5_ components and diabetes mortality. A study in eight European cohorts reported that BC was associated with diabetes mortality (HR 1.24, 95% CI: 1.11–1.38) [39]. However, the BC measured in their study was not the only part of the PM_2.5_ components.

Our study advances the existing literature by providing comprehensive evidence of how various PM_2.5_ components uniquely contribute to mortality linked with IS, COPD and DM. This differentiation is vital as it enhances our understanding and supports the development of specific environmental and public health interventions. This highlights the necessity for further research into their intricate relationships with public health. For the benefit of comparative analysis in future studies, it would be advantageous if similar methods for estimating exposure were adopted and the same components were examined. [21,40].

In this study, we also identified susceptible populations by stratified analyses and the susceptible population varied deaths caused by specific diseases. Our findings of the varied effects of OM and BC on mortality risks by sex and age (when exposed to higher levels of OM and BC, males and the younger population have increased diabetes mortality risk, while females and the elderly population have increased COPD mortality risk) were plausible and can be explained by existing studies. Due to the worse lung function in the elderly population, particulate matter can affect them more severely and thus cause more death events [20]. Previous studies showed that females have a poorer prognosis and higher mortality rate for COPD than males [41]. Studies have also shown that the effects of PM_2.5_ on daily mortality from respiratory diseases were stronger among females than males [42]. Our study further revealed that this higher mortality rate observed in females could be driven by higher exposure to OM and BC in terms of PM_2.5_. The difference in physiological structure in different populations, such as the size of airway diameter, genetic factors and hormonal level, might cause these various effects.

Additionally, we found a linear exposure–response relationship without a threshold between PM_2.5_ components and IS, COPD and DM mortality. This finding aligns with the recently proven evidence that there is no safe level of air pollution in terms of health effects [43]. Similar to another study, we found that an increase in concentrations was associated with an increased risk of mortality [44].

Our study included all of the population in Beijing, China, which was a large sample size of participants, yielding a large statistical power and a good generalizability for the population in Beijing. We used a multi-pollutant model that considered other pollutants known to be related to mortality and also adjusted for meteorological factors, which will minimize the bias. Our study is, to our knowledge, the first to investigate the associations between the five most abundant PM_2.5_ components and mortality from IS, COPD and DM—three important chronic diseases that account for a large burden for premature deaths.

This research also has several limitations. First, there are potential residual confounders that were unable to be adjusted or considered in our models. The absence of data on other pollutants such as O_3_ and CO could potentially bias the associations we found. These pollutants are known to have health impacts that could overlap with those observed in our study and thus might confound our findings. Other residual confounders such as mortality-related behaviors including cigarette smoking, alcohol consumption or physical activity could also bias the results. Future studies should aim to include these pollutants and factors to provide a more comprehensive understanding of the air quality–health nexus. Second, the selection of these five PM_2.5_ components could lead to a selection bias. Although these five components assessed in our study accounted predominately for the total PM_2.5_ concentration, other components still accounted for 17.5% of the PM_2.5_ concentration. These components include various toxic substances such as lead, arsenic, benzene, chloride, formaldehyde, manganese, nickel and polycyclic organic matter, which can bias the results. Third, we did not consider personal air pollutant exposure levels affected by daily activity patterns or indoor air pollution due to the ecological investigation of this study. Misclassification of the exposure could bias our findings. Last, the data used in this study were based in Beijing, which had limited generalizability to other regions in China or other countries. Further studies are needed to replicate the observed associations found in our study.

In line with previous studies, we found that cumulative lag effects on mortality were generally higher than the single-day effects [14]. Achilleos et al. reported in their systematic review and meta-analysis that the mortality effect of PM_2.5_ for a single-day exposure was lower than the two-day average exposure. This finding indicates that studies investigating single-day exposure to PM_2.5_ components might underestimate their health impacts. It is therefore important for future studies to consider multiple lag days in their analysis when investigating the acute effects of PM_2.5_ components.

Identification of the vulnerable groups highlighted the necessity of tailored strategies in preventing premature deaths caused by these air pollutants. The results from this study and future similar studies can be used to help formulate more detailed air quality criteria covering PM_2.5_ components for susceptible populations with chronic diseases.

## 5. Conclusions

In conclusion, we found evidence that short-term exposure to PM_2.5_ components was associated with IS, COPD and DM mortality. Cumulative higher exposure to SO_4_^2−^ and NH_4_^+^ was associated with an increased risk of IS mortality, while cumulative higher exposure to OM and BC was associated with an increased risk of COPD and DM mortality. These different short-term effects of PM_2.5_ components found in this study provided evidence for the differences found in studies investigating PM_2.5_ as a whole. We found that these effects also varied by sex and age group, which highlighted the vulnerable groups for specific components and mortality risks. Our findings contribute additional scientific evidence of the association between specific PM_2.5_ components and health outcomes, which could inform the formulation of regulatory policies and air quality standards. However, it is important to clarify that these associations do not imply causality. Further studies are essential to consolidate the transition from associative to causal relationships, thereby providing a more robust basis for public health interventions and policy formulation.

## Figures and Tables

**Figure 1 toxics-12-00381-f001:**
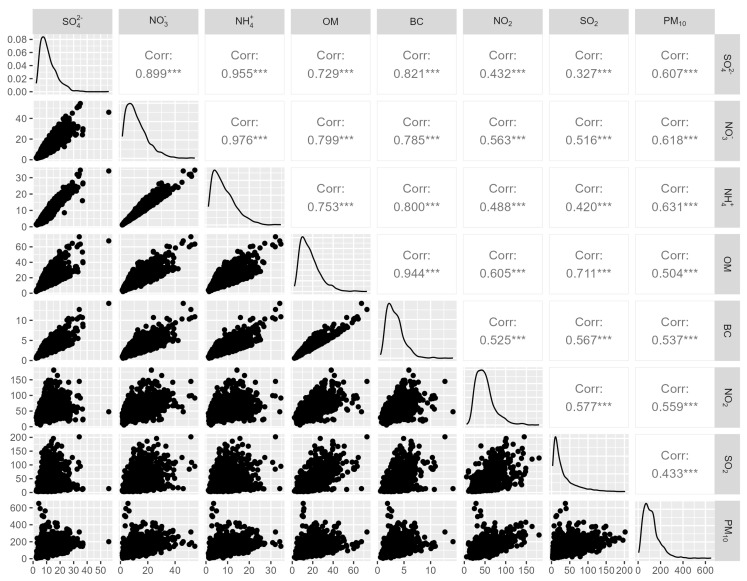
Spearman’s correlation coefficients between daily air pollutant concentrations. Abbreviations: BC: black carbon; Corr: correlation coefficients; NH_4_^+^: ammonium ion; NO_2_: nitrogen dioxide; NO_3_^−^: nitrate ion; OM: organic matter; PM_10_: fine particulate matter less than 10 μm in aerodynamic diameter; SO_2_: sulfur dioxide; SO_4_^2−^: sulfate ion; ***: indicates statistical significance with *p* < 0.001.

**Figure 2 toxics-12-00381-f002:**
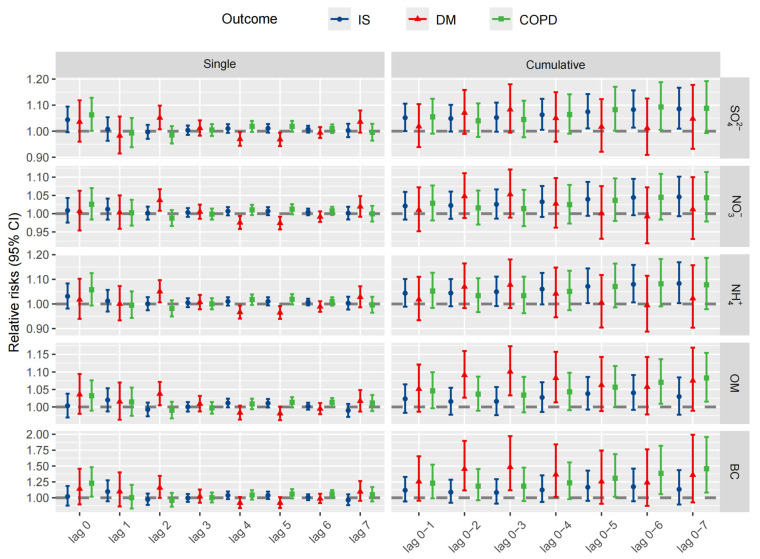
Relative risks (95% CI) for single−day lag and cumulative−day lag for IS, DM and COPD mortality. Abbreviations: BC: black carbon; CI: confidence interval; COPD: chronic obstructive pulmonary disease; DM: diabetes mellitus; IS: ischemic stroke; NH_4_^+^: ammonium ion; NO_3_^−^: nitrate ion; OM: organic matter; SO_4_^2−^: sulfate ion.

**Figure 3 toxics-12-00381-f003:**
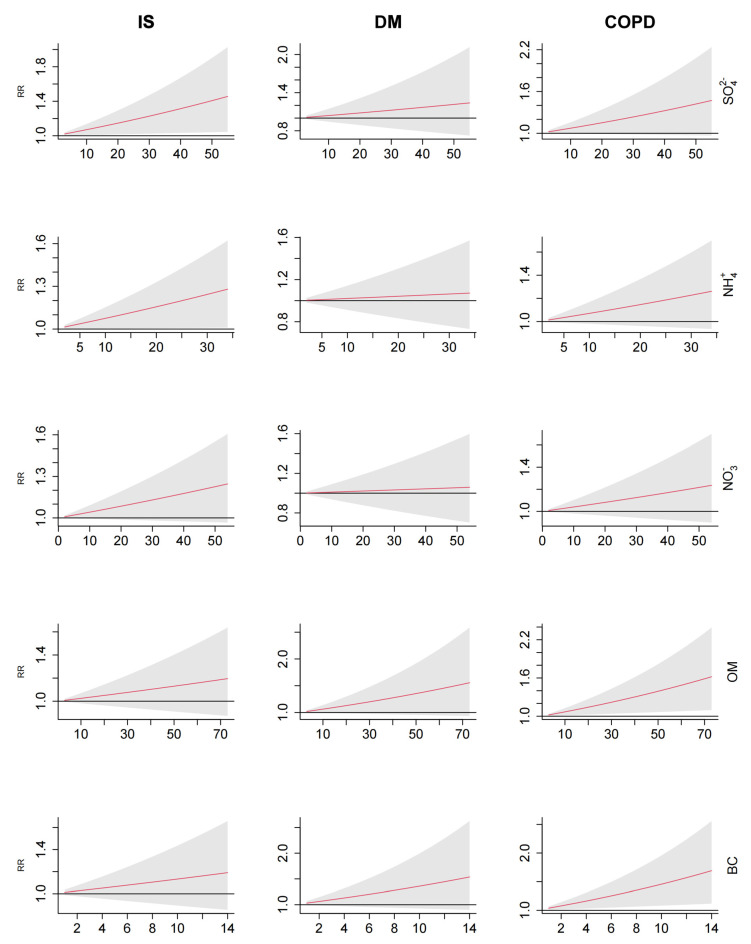
Exposure−response relationships for SO_4_^2−^, NH_4_^+^, NO_3_^−^, OM and BC (lag 0−7 days) with IS, DM and COPD mortality. Abbreviations: BC: black carbon; COPD: chronic obstructive pulmonary disease; DM: diabetes mellitus; IS: ischemic stroke; NH_4_^+^: ammonium ion; NO_3_^−^: nitrate ion; OM: organic matter; RR: Relative risks; SO_4_^2−^: sulfate ion.

**Table 1 toxics-12-00381-t001:** Descriptive summary of daily death numbers, air pollution concentrations and weather conditions in Beijing, China (2008–2011).

Variable	Mean	SD	MAD	Min	P_25_	P_50_	P_75_	IQR	Max
Daily death counts									
IS	15	5	4	4	11	14	18	7	36
COPD	9	4	3	1	6	8	11	5	27
DM	5	2	3	0	4	5	7	3	14
PM_2.5_ components (μg/m^3^)									
SO_4_^2−^	11.24	6.02	5.28	2.34	6.73	9.79	14.26	7.53	55.98
NH_4_^+^	8.37	5.30	5.03	1.10	4.18	7.19	11.31	7.13	34.62
NO_3_^−^	12.00	8.00	7.11	1.34	6.01	10.08	16.24	10.23	54.44
OM	17.34	9.90	8.87	2.09	10.02	15.31	22.35	12.33	73.06
BC	3.32	1.60	1.50	0.58	2.12	3.06	4.18	2.06	14.21
Other air pollutants (μg/m^3^)									
NO_2_	49.82	23.13	19.42	5.37	33.17	45.90	60.00	26.83	180.67
SO_2_	32.27	31.66	16.31	3.00	10.96	20.00	42.00	31.04	202.00
PM_10_	117.75	74.14	60.31	4.91	65.09	104.00	146.55	81.46	651.18
Meteorological factors									
Temp (°C)	13.21	11.34	14.97	−12.50	2.20	14.90	24.00	21.8	34.50
Humd (%)	50.87	19.97	25.20	9.00	34.00	52.00	67.00	33	95.00
Pressure (hPa)	1012.42	10.28	12.16	989.70	1004.10	1011.80	1020.50	16.4	1039.30

Abbreviations: BC: black carbon; COPD: chronic obstructive pulmonary disease; DM: diabetes mellitus; MAD: mean absolute deviation; Max: maximum, the highest value in the data set; Min: minimum value, the lowest value in the data set; NH_4_^+^: ammonium ion; NO_2_: nitrogen dioxide; NO_3_^−^: nitrate ion; OM: organic matter; P_25_: 25th Percentile, also known as the first quartile, indicating the value below which 25% of the data falls; P_50_: 50th Percentile, also known as the median, marking the middle value of the data set; P_75_: 75th Percentile, also known as the third quartile, indicating the value below which 75% of the data falls; PM_10_: particulate matter less than 10 μm in aerodynamic diameter; IQR: interquartile range, calculated as P_75_ minus P_25_, representing the middle 50% of the data; IS: ischemic stroke; SD: standard deviation; SO_2_: sulfur dioxide; SO_4_^2−^: sulfate ion.

**Table 2 toxics-12-00381-t002:** Attributable fractions and numbers of IS, COPD and DM mortality attributable to SO_4_^2−^, NH_4_^+^, NO_3_^−^, OM and BC.

Air Pollution	IS		COPD		DM	
	AN	AF (95% CI)	AN	AF (95% CI)	AN	AF (95% CI)
SO_4_^2−^	1597	7.36 (1.03, 13.59)	1004	7.48 (−0.94, 14.63)	345	4.34 (−6.93, 15.02)
NH_4_^+^	1279	5.90 (0.29, 11.52)	741	5.53 (−1.78, 12.04)	142	1.79 (−8.09, 10.19)
NO_3_^−^	1047	4.83 (−0.52, 9.97)	625	4.66 (−1.94, 10.65)	111	1.40 (−8.77, 9.64)
OM	938	4.32 (−2.58, 10.91)	1528	11.39 (1.80, 19.65)	814	10.24 (−1.63, 20.69)
BC	900	4.15 (−3.95, 12.11)	1611	12.01 (2.16, 20.42)	783	9.85 (−2.78, 20.50)

Abbreviations: AF: attributable fractions; AN: attributable numbers; BC: black carbon; COPD: chronic obstructive pulmonary disease; CI: confidence interval; DM: diabetes mellitus; NH_4_^+^: ammonium ion; NO_3_^−^: nitrate ion; OM: organic matter; IS: ischemic stroke; SD: standard deviation; SO_2_: sulfur dioxide; SO_4_^2−^: sulfate ion.

## Data Availability

The data presented in this study are available upon request from the corresponding author.

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
