# Peer review of "Associations between PM2.5 Components and Mortality of Ischemic Stroke, Chronic Obstructive Pulmonary Disease and Diabetes in Beijing, China"

_toxics, 2024, doi:10.3390/toxics12060381_

Round 1
Reviewer 1 Report
Comments and Suggestions for Authors
This manuscript describes a study that investigates associations between different PM2.5 components and mortality resulting from ischemic stroke, chronic obstructive pulmonary disease and diabetes in Beijing, China. Using data collected from a Tracking Air Pollution database and daily mortality data obtained from the Chinese Centre for Disease control and Pollution, they report associations between mortality and short term exposure to different PM2.5 components with different lag periods. The study, like many others, uses available data collected by other sources and has no specific hypothesis to test in terms of lag times – ie it is hypothesis generating – but given the frequency of such papers (and often the contradictory nature of results obtained) there are limitations in the interpretability of the data obtained. The study design could have been improved by having two arms to it: one a hypothesis generating and the second a hypothesis testing on a different population. Results from such studies would then be more convincing. Why did the authors choose this study design?
As the authors indicate there are several limitations to this study including the lack of any personal air exposure estimates. Indeed, it is unclear how exposures for the study population were actually estimated. Did they estimated exposure based upon residential address of the person, place where the person died or just a single measure of pollution in Beijing on the day in question. There may be significant geographical variation in pollution levels which may not have been accounted for in this study. Please clarify how exposure was estimated.
A significant number of statistical tests have been carried out and, of course, some of these will occur by chance. The authors do not seem to have taken this into account either in their analysis or in the description of the results. Nor indeed whether their results are biologically meaningful. For example, in the abstract the authors report “After adjusting for temperature, 30 relative humidity, pressure, SO2, NO2 and PM10, an IQR increase of OM at lag day 2 and lag day 6 31 was associated with an increased DM mortality risk (RR 1.038; 95% CI: 1.005-1.071) and COPD mor-32 tality risk (RR 1.013; 95% CI: 1.001-1.026)” Is there any evidence that effects on lag day 2 and lag day 6 are biologically meaningful?
Comments on the Quality of English Language
Minor English editing is required.
Reviewer 2 Report
Comments and Suggestions for Authors
This study identifies significant associations between short-term exposure to specific PM2.5 components and increased mortality from ischemic stroke, chronic obstructive pulmonary disease, and diabetes. It is interesting, however it need some corrections.
Suggestion: I would not split between lines words “is-chemic” and “dia-betes”
Line 23: Please explain PM2.5
Line 34 and line 64, and 175: SO2, NO2 and PM10 – change into subscript.
Table 1: Explain all the used abbreviations, e.g. SD, P25, P50, P75, IQR, Max
Figure 1: Explain all the used abbreviations. Please use the constant expressions, e.g. if there is in the text “SO2” it should be the same on the figure, and we see “so2”. Please change all.
Figure 2 and 3: Please provide a better-quality figure with a bigger font. Now, it is impossible to read. A figure like this CAN NOT be published—it’s not a figure with data; it is an art drawing now.
Table 2: What is AN, AF?
Line 191: “The results of single-day lag (0-7) and cumulative lags (01-07)” – I think it should be “…and cumulative lags (1-7).” In general, there is a lot of inconsistent in the text. Please go through the text very carefully and modify those; it loses professionalism.
I’m missing a figure or a graph showing an association between PM and mortality (in my opinion, it is a must-be). It would be much more interesting for the readers.
Reviewer 3 Report
Comments and Suggestions for Authors
See attached file for comments.

Comments on the Quality of English LanguagePlease perform a complete check of the English text.
Round 2
Reviewer 2 Report
Comments and Suggestions for Authors
The authors addressed all comments.
Reviewer 3 Report
Comments and Suggestions for Authors
I thank and congratulate the authors for having addressed all my notes precisely and comprehenisively, adapting the manuscript accordingly.
The paper seems now ready for publication.